# Bone Metastases of Diverse Primary Origin Frequently Express the VDR (Vitamin D Receptor) and CYP24A1

**DOI:** 10.3390/jcm11216537

**Published:** 2022-11-03

**Authors:** Jonas Seiler, Regina Ebert, Maximilian Rudert, Marietta Herrmann, Ellen Leich, Manuela Weißenberger, Konstantin Horas

**Affiliations:** 1Department of Musculoskeletal Tissue Regeneration, Koenig-Ludwig-Haus, University of Wuerzburg, 97074 Wuerzburg, Germany; 2Department of Orthopaedic Surgery, Koenig-Ludwig-Haus, University of Wuerzburg, 97074 Wuerzburg, Germany; 3IZKF Research Group Tissue Regeneration in Musculoskeletal Diseases, University Clinics Würzburg, 97070 Wuerzburg, Germany; 4Orthopaedic Center for Musculoskeletal Research, University of Würzburg, 97074 Wuerzburg, Germany; 5Department of Pathology, University of Wuerzburg, 97070 Wuerzburg, Germany; 6Comprehensive Cancer Center Mainfranken, University Hospital of Wuerzburg, 97080 Wuerzburg, Germany; 7Department of Pathology, Caritas Hospital Bad Mergentheim, 97980 Bad Mergentheim, Germany

**Keywords:** vitamin D receptor, VDR, CYP24A1, bone metastasis, vitamin D

## Abstract

Active vitamin D (1,25(OH)2D3) is known to exert direct anti-cancer actions on various malignant tissues through binding to the vitamin D receptor (VDR). These effects have been demonstrated in breast, prostate, renal and thyroid cancers, which all have a high propensity to metastasise to bone. In addition, there is evidence that vitamin D catabolism via 24-hydroxylase (CYP24A1) is altered in tumour cells, thus, reducing local active vitamin D levels in cancer cells. The aim of this study was to assess VDR and CYP24A1 expression in various types of bone metastases by using immunohistochemistry. Overall, a high total VDR protein expression was detected in 59% of cases (39/66). There was a non-significant trend of high-grade tumours towards the low nuclear VDR expression (*p* = 0.07). Notably, patients with further distant metastases had a reduced nuclear VDR expression (*p* = 0.03). Furthermore, a high CYP24A1 expression was detected in 59% (39/66) of bone metastases. There was a significant positive correlation between nuclear VDR and CYP24A1 expression (*p* = 0.001). Collectively, the VDR and CYP24A1 were widely expressed in a multitude of bone metastases, pointing to a potential role of vitamin D signalling in cancer progression. This is of high clinical relevance, as vitamin D deficiency is frequent in patients with bone metastases.

## 1. Introduction

Cancer is a leading cause of death worldwide, accounting for almost 10 million deaths in 2020, according to the World Health Organization [1]. During cancer progression, it is estimated that over 50% of patients are predicted to develop metastases [2]. Bone metastases are among the most frequent sites of metastasis spread of late-stage cancers [3]. Although tumour therapy has improved significantly over recent years, the treatment of patients with bone metastases is mainly restricted to palliative measures. Moreover, metastasis to bone noticeably impairs the patient’s quality of life [3,4].

The active form of vitamin D (1,25(OH)_2_D_3_) is known to exert growth inhibitory and pro-differentiating effects on many different cancer cells [5,6,7]. 25OHD_3_ is the biologically inactive 1,25(OH)_2_D_3_ precursor and a reliable and robust marker of vitamin D status, which is catalysed to 1,25(OH)_2_D_3_ via 1α-hydroxylase (CYP27B1). Consequently, several studies have provided substantial evidence for the protective role of 1,25(OH)_2_D_3_, as increased circulating levels of 25OH vitamin D (25OHD_3_) decrease the risk of, e.g., breast cancer development and progression [8,9]. In addition, it has been demonstrated that supplementation with vitamin D_3_ reduces the incidence of advanced (metastatic or fatal) cancer in a large randomised controlled trial, specifically designed to examine cancer risk [10]. In animal models, it has been shown that vitamin D deficiency promotes cancer growth in bone [11,12]. On the one hand, this is attributable to indirect effects due to increased bone remodelling and changes in the bone microenvironment as a result of vitamin D_3_ deficiency [13,14]. On the other hand, this can be explained by the absence of the known direct anti-proliferative, pro-apoptotic and pro-differentiation effects of 1,25(OH)_2_D_3_ on cancer cells [5,15,16]. These direct vitamin D anti-cancer actions are mainly mediated through the binding of 1,25(OH)_2_D_3_ to the vitamin D receptor (VDR).

Interestingly, some studies suggest that the VDR itself possesses a ligand-independent function affecting cancer growth and the metastatic potential [17,18,19]. Previously, we reported that the loss of the VDR promotes an epithelial to mesenchymal transition, which increased cell invasiveness and metastasis to bone in a murine model of breast cancer metastasis to bone [18]. Consistent with these results, a low VDR expression has been demonstrated to increase the metastatic potential in melanoma, colorectal and urothelial cancer cells [20,21,22]. Additionally, a low VDR expression in various primary tumours has been associated with aggressive tumour characteristics and poor differentiation, suggesting that VDR-mediated 1,25(OH)_2_D_3_ actions are dysregulated in cancer cells [23,24,25]. Furthermore, the downregulation of the VDR has been correlated with worse clinical outcomes in, e.g., breast, prostate and colon cancer patients [21,24,25]. However, other studies have indicated no significant association between a low VDR expression and poor clinical outcomes [26].

The binding of 1,25(OH)_2_D_3_ to the VDR increases the expression of the vitamin D catabolising enzyme, CYP24A1 (24-hydroxylase), which, in turn, results in a negative feedback mechanism. In addition to regulating the circulating concentrations of 1,25(OH)_2_D_3_, CYP24A1 may also reduce the levels of 1,25(OH)_2_D_3_ within cells [27]. Intriguingly, CYP24A1 expression has been found to be significantly higher in malignant tumours of prostate, lung and colorectal cancers compared to benign tumours [28,29,30]. Further, several studies have demonstrated that the increased expression of CYP24A1 may cause resistance to vitamin D anti-cancer actions [28,31,32]. Moreover, the inhibition of CYP24A1 in 1,25(OH)_2_D_3_-insensitive cancer cells enhances vitamin D anti-proliferative effects [33,34].

Collectively, a low VDR expression and the alteration of vitamin D catabolism may contribute to tumour progression and the metastatic process. A low VDR expression in primary tumours is correlated with the metastatic spread of tumour cells. However, there are currently no reports of studies investigating VDR expression and vitamin D catabolism in bone metastases. For this reason, the purpose of this study was to analyse VDR and CYP24A1 expression in bone metastases of different primary origin. In addition, our aim was to investigate possible associations of VDR and CYP24A1 expression and tumour characteristics, as well as patient outcomes.

## 2. Materials and Methods

### 2.1. Sample Collection

Altogether, tissue samples of bone metastases were obtained from 66 cancer patients. Samples were previously collected directly either from tumour biopsies or tumour resections conducted at the University Hospital of Wuerzburg, Germany. For sample processing, all samples were provided as paraffin-embedded tissue samples by the Department of Pathology, University of Wuerzburg, Germany.

The study was conducted in accordance with the guidelines of the local committee of medical ethics (ethics number 146/16-MK) and in accordance with the World Medical Association Declaration of Helsinki.

### 2.2. Sample Processing

Paraffin-embedded sections were cut from bone metastases using a microtome (Leica SM 2000r, Nussloch, Germany). Sections of 3 μm thickness were generated, decalcified, placed on silanated slides (SuperFrost^®^Plus, R. Langenbrinck, Emmendingen, Germany) and heat cured for an hour at 56 °C.

### 2.3. Immunohistochemistry (IHC)

Paraffin sections were re-hydrated through the use of xylol (Carl Roth GmbH + Co. KG, Karlsruhe, Germany), a series of decreasing concentrations of ethanol (Carl Roth GmbH + Co. KG, Karlsruhe, Germany) (96% ethanol, 80% ethanol, 60% ethanol for 3 min each) and rinsed with water for 3 min. Subsequently, heat-induced antigen retrieval followed using a water bath at 95 °C with 0.01 M citrate buffer (Agilent Dako, Santa Clara, CA, USA) (pH 6.1) and a cooling down period. Afterwards, the paraffin sections were water-cleared and an endogenous peroxidase block was applied using 3% H2O2 (Merck KGaA, Darmstadt, Germany). The VDR and CYP24A1 IHC was performed on different samples from the same metastasis. The IHC was conducted using a validated rat monoclonal anti-body for VDR (DLN-013017; Dianova, Hamburg, Germany) in a 1:100 dilution (S202230-2, Agilent Technologies, Santa Clara, CA, USA) to assess the VDR expression. CYP24A1 was assessed using the IHC-validated rabbit poly-clonal anti-body for CYP24A1 (HPA022261-100UL, Sigma Aldrich, St. Louis, MO, USA) in a 1:200 dilution (S202230-2, Agilent Technologies, Santa Clara, CA, USA). Afterwards, sections were incubated for 60 min at room temperature and cleared with phosphate-buffered saline (PBS-powder, AppliChem GmbH, Darmstadt, Germany). Subsequently, the secondary anti-body for the VDR IHC (SuperVision-Set 2, HPR Single Species Rat, DCS, Hamburg, Germany) and CYP24A1 IHC (Supervision-Set 2, HRP Single Species Rabbit, DCS, Hamburg, Germany) was applied, respectively. Initially, the polymer enhancer was applied and incubated for 30 min, followed by the application and a 30 min incubation of the second component, the polymer reagent. After being rinsed with PBS again, the sections were developed with diaminobenzidine (DAB; Dako Liquid DAB + Substrate Chromogen System, Agilent Technologies, Santa Clara, CA, USA) and then counterstained with a haematoxylin solution (Carl Roth GmbH + Co. KG, Karlsruhe, Germany). Finally, the paraffin sections were dehydrated using ascending concentrations of ethanol (50%, 70%, 96% for three minutes each), isopropanol (AppliChem GmbH, Darmstadt, Germany) and xylol and sealed.

Positive and negative controls were generated and included in the staining and evaluation process. For this purpose, healthy kidney tissue served as a positive control, which was stained according to the protocol used for the bone metastases. Negative controls using rabbit serum (CYP24A1; Sigma Aldrich, St. Louis, MO, USA) or rat serum (VDR; Sigma Aldrich, St. Louis, MO, USA) instead of the respective primary anti-bodies were used to ensure the specific binding to the respective VDR/CYP24A1 epitope. In particular, the serum was diluted in PBS to the same protein concentration as the CYP24A1/VDR-antibody dilution. Afterwards, the serum–PBS mixture was applied in the same dilution (1:200 CYP24A1; 1:100 VDR). Subsequently, the same staining protocol as for CYP24A1/VDR was performed.

### 2.4. Evaluation

The VDR and CYP24A1 staining of bone metastasis samples was analysed for VDR or CYP24A1 expression using light microscopy (Axio Observer 7, Carl Zeiss Microscopy GmbH, Oberkochen, Germany). The VDR and CYP24A1 expression was graded following an established scoring system (e.g., similarly used by [21,26]) (immunoreactive score by Remmele and Stegner (IRS)) containing the assessment for the intensity of VDR staining (0 to 3) and the percentage of tumour cells stained within the scoring region (0 = 0%; 1 ≤ 10%; 2 = 10% to 50%; 3 = 51% to 80%; 4 ≥ 80%). All samples were scored independently and blindly by three observers (JS, MW and KH). All scores were multiplied with each other (0–12 points) and, afterwards, the three observer’s average value was calculated (for further details see the Appendix A). Subsequently, patients were dichotomised into two (high vs. low) groups of bone metastases protein expression. The threshold value was set at an IRS of 5.

### 2.5. Statistical Evaluation/Data Analysis

A statistical analysis was conducted using SPSS version 27 (IBM Corporation, Armonk, NY, USA). Patient data were collected and analysed (Table 1). Firstly, the Shapiro–Wilk test was used to test for Gaussian distribution. The expression of VDR and CYP24A1 was then correlated to histopathological grading (grades 1 to 3) and TNM stages using Pearson’s chi-square test. Furthermore, the Mann–Whitney U test was used to compare the characteristics of the patient groups (Table 2). Differences between groups were considered statistically significant when *p* < 0.05. The Spearman Rho correlation coefficient was used to evaluate the correlation between the VDR and CYP24A1 expression.

## 3. Results

### 3.1. Baseline Characteristics of Participants

In this study, a total of 66 patients, including 35 (53%) females and 31 (47%) males, with bone metastases of different primary origin, was enrolled. Bone metastases were secondary to prostate, breast, kidney, lung, follicular thyroid, gastro-intestinal and other cancers (see Table 1 for details). The mean age of primary cancer diagnosis was 61.81 years, while bone metastases occurred at a mean age of 64.45 years. Thus, bone metastases were diagnosed 2.64 years on average after the primary cancer diagnosis. The median survival after bone metastasis diagnosis was 18 months (95% CI 9.3–26.6 months) (*n* = 34; some patients’ survival data were missing). Most bone metastases were located in the femur, accounting for 34 (52%) patients. Other locations were the vertebral bodies (17%), humerus (9%), cranium (8%) and pelvis (6%). Less frequent were locations such as the radius or scapula (9% in total). Overall, 36 patients (49%) had lymph node metastasis (N+), 16 had no lymph node involvement (N0) and 18 patients had an unknown lymph node status. In total, 20 (30%) patients presented with poorly differentiated cancers (G3), 28 (42%) with G2 cancers and G1 cancers were identified in five patients (8%). Moreover, 31 (47%) patients had extra-osseous metastases in addition to bone metastasis (see Table 1 for further cancer characteristics).

### 3.2. VDR Protein Expression in Bone Metastases

Staining for the VDR in bone metastases was evident in the nucleus and cytoplasm, and was evaluated separately. Additionally, the total VDR protein expression was assessed. A high VDR expression in total was detected in 39/66 (59%) specimens. Respectively, a high VDR nuclear protein expression was identified in 47/66 (71%). Notably, all prostate cancer bone metastases (*n* = 8) demonstrated a high nuclear VDR protein expression in this study cohort. In comparison, breast (*n* = 16) and lung (*n* = 11) cancer bone metastases showed rather low nuclear and total VDR expressions (Figure 1B). Furthermore, the cytoplasmic VDR expression was evaluated, whereby a high cytoplasmic VDR expression was detected in 37/66 (56%) of bone metastases. Interestingly, prostate cancer bone metastases had a low VDR protein expression in the cytoplasm more often compared to the nucleus. Other primary cancer metastases to bone demonstrated a similar expression of the nucleus and cytoplasm (Figure 1B).

Statistical analyses comparing the VDR expression and lymph node involvement (N0 vs. N+) did not show any significant differences. Additionally, the sub-group analysis of the various primary cancers revealed no correlation in the primary cancer’s infiltration depth (T-stage) with VDR expression. However, there was a non-significant trend in high-grade cancers towards low nuclear VDR expressions in bone metastases (Figure 1C and Table 2). Interestingly, patients with further metastases other than bone metastases had reduced nuclear VDR levels compared to patients without other distant metastases (Figure 1C).

### 3.3. CYP24A1 Protein Expression in Bone Metastases

Since the 1,25(OH)2D3-bound VDR acted as a transcription factor for CYP24A1, the correlation between the VDR and CYP24A1 expression was analysed. Here, a positive significant correlation between the VDR expression and CYP24A1 was observed (*p* < 0.001).

CYP24A1 was clearly identified in the cytoplasm. A high CYP24A1 expression was detected in 39/66 (59%) of bone metastases. Gastro-intestinal (*n* = 9) and prostate cancer bone metastases had a tendency towards having a high CYP24A1 protein expression, whereas, in particular, breast cancer bone metastases had a rather low CYP24A1 protein expression (Figure 2B).

Statistical analyses correlating bone metastases, CYP24A1 protein expression and TNM stages (*TNM Classification of Malignant Tumours*) did not show any significant differences (Table 2). Furthermore, no significant difference in CYP24A1 protein expression was found between well and poorly differentiated cancers. Similar to the VDR expression, CYP24A1 expression was significantly decreased in patients with multiple metastatic cancers (Figure 2B).

## 4. Discussion

Our study demonstrated a widely distributed VDR expression in bone metastases secondary to breast, prostate, renal, gastro-intestinal, follicular thyroid and further cancers (Table 1). To the best of our knowledge, there have not yet been any reports of studies investigating the VDR expression in bone metastases. However, VDR expression had been evaluated in healthy tissues, as well as diverse primary cancer cells in various studies [35,36]. In particular, primary pancreas, prostate, breast and colorectal cancer cells demonstrated a reduced VDR expression compared to benign tissues, implying that VDR expression is dysregulated in cancer cells [23,37,38,39]. Moreover, in bone metastases, we found a non-significant trend in poor differentiated tumours towards low nuclear VDR expressions. In line with these results are studies investigating the VDR expression in primary cancers, such as, e.g., in prostate, breast and pancreas cancers, a low VDR expression has been associated with aggressive cancer characteristics and poor differentiation [23,24,25]. A high VDR expression in contrast has been linked to lower mortality in primary breast, prostate, lung and colon cancers [21,24,25,40]. Thus, a low VDR expression may be a prognostic marker for cancer development and progression. Additionally, patients with multiple metastatic cancers had a significantly lower bone metastatic nuclear VDR expression in this study. In clinical studies of primary cancers, a low VDR expression was associated with metastases in colon, urothelial and breast cancers [21,22,41]. Likewise, we previously demonstrated in a murine model that the loss of the VDR resulted in a significantly greater cell invasiveness and skeletal tumour burden [18]. Similarly, Zhang et al. demonstrated in vivo that breast cancers with a high VDR expression could metastasise significantly less to the lungs [42].

In pre-clinical models, growth-reducing effects of 1,25(OH)_2_D_3_ application have been shown in prostate [15], breast [43], lung [16], kidney [44] and, also, thyroid cancers [45]. In breast cancer cells, it has been demonstrated that these effects depend on the nuclear VDR expression [46]. Therefore, the nuclear VDR expression of cancer cells may be fundamental for potential 1,25(OH)_2_D_3_ anti-cancer effects. In our study, almost all types of bone metastases had a similar VDR expression of the nucleus compared to the cytoplasm. However, prostate cancer bone metastases (*n* = 8) demonstrated a high nuclear VDR expression compared to the cytoplasm more frequently. As our study indicated that the VDR was widely distributed in different types of bone metastases, patients with bone metastases may benefit from adequate vitamin D levels. Additionally, vitamin D_3_ deficiency increases bone remodelling, which has been suggested to increase the risk of bone metastasis [11,12]. Consequently, vitamin D supplementation is likely to be beneficial, as 25OHD_3_ deficiency is common in patients with bone metastases [47,48]. Moreover, vitamin D_3_ supplementation could reduce the risk of metastatic or fatal cancers [10,49].

However, the measured serum 25OHD_3_ levels could not be equivalent to the cancer cells’ microenvironment local 1,25(OH)_2_D_3_ level. Indeed, local enzymes (CYP24A1 as the catabolising enzyme and CYP27B1 as the activating enzyme) may affect the local 1,25(OH)_2_D_3_ level. In healthy tissues, these enzymes are tightly regulated depending on the availability of 1,25(OH)_2_D_3_. Interestingly, studies suggest that CYP24A1 is increased in malignant tumour cells compared to healthy tissue, leading to an enhanced 1,25(OH)_2_D_3_ de-activation [28,30,50]. In this study, we identified the local expression of the 1,25(OH)_2_D_3_ catabolising enzyme CYP24A1 in bone metastases. Thus far, there have not been any reports on CYP24A1 studies in bone metastases. Comparatively, in primary cancers, a high CYP24A1 expression has been associated with poor cancer cell differentiation in prostate cancer [28] and a poor clinical outcome in lung and colon cancers [30,50]. In contrast, breast cancer patients with low-CYP24A1-expressing cancers had a reduced overall survival [51]. In bone metastases, CYP24A1 and VDR expression was reduced in patients with multiple metastatic tumours. Poorly differentiated tumours demonstrated a similar CYP24A1 protein expression compared to well differentiated cancers. This did not confirm our hypothesis of increased vitamin D catabolism in aggressive tumours, thereby reducing local 1,25(OH)_2_D_3_ levels. However, VDR and CYP24A1 protein levels correlated significantly. In healthy tissues, the activated VDR enhanced the transcription of CYP24A1 genes in the sense of a negative feedback mechanism. A similar association might be possible in cancer cells. Therefore, it is possible that the entire vitamin D–VDR axis dysregulation in cancer cells may contribute to the metastatic process. For example, it is feasible that tumour cells express less CYP24A1 as a result of a dysregulated and low VDR expression. Consequently, the VDR-mediated transcription of 1,25(OH)_2_D_3_-sensitive genes, which confer vitamin D anti-cancer effects, was reduced. Therefore, further studies are needed to determine a possible increased expression of CYP24A1 in relation to the VDR expression. Furthermore, it would be interesting to analyse the expression of CYP27B1 and its interactions with VDR and CYP24A1 in cancer cells. This should certainly be regarded as a limitation of the study, as the vitamin-D-activating enzyme CYP27B1, the patients’ vitamin D levels, possible vitamin D supplementation and PTH levels were not recorded. Therefore, we could not know the systemic VDR ligand concentrations. However, these would not provide us with any information about the local 1,25(OH)_2_D_3_ levels. Moreover, an analysis of the VDR and CYP24A1 expression of circulating disseminated cancer cells and the corresponding primary cancer tissues would have been interesting for studying the metastatic process more precisely. Further study limitations were that the study size was modest, since metastatic tissue can only be obtained on rare occasions, limiting the significance of our findings. Moreover, patient data were not complete, therefore, reducing the significance of the statistical evaluation.

Collectively, our study indicated that the VDR and CYP24A1 were broadly expressed in bone metastases of breast, prostate, renal, gastro-intestinal, follicular thyroid and other cancers. Thus, patients with bone metastases and, in particular, with vitamin D deficiency, may benefit from vitamin D supplementation, as it is likely to impair the indirect and direct effects of metastatic growth. Furthermore, a low VDR expression and altered vitamin D metabolism reduced the vitamin D anti-cancer effects. Consequently, it is possible that low-VDR-expressing cancer cells may be able to evade the VDR-mediated anti-cancer characteristics of vitamin D. Therefore, further large studies investigating all aspects of the vitamin D metabolism and signalling in primary cancers and in the corresponding metastases are needed.

## 5. Conclusions

Our findings provided evidence that the VDR and CYP24A1 are widely expressed in bone metastases of different primary origin. Additionally, we found that poorly differentiated tumours had a trend towards having a low VDR expression. As vitamin D is known to directly suppress tumour growth via the VDR, adequate vitamin D levels are of utmost importance for patients with bone metastases. However, vitamin D deficiency is frequent in patients with bone metastases. As such, vitamin D supplementation might be of importance, especially for patients with VDR-expressing bone metastases.

## Figures and Tables

**Figure 1 jcm-11-06537-f001:**
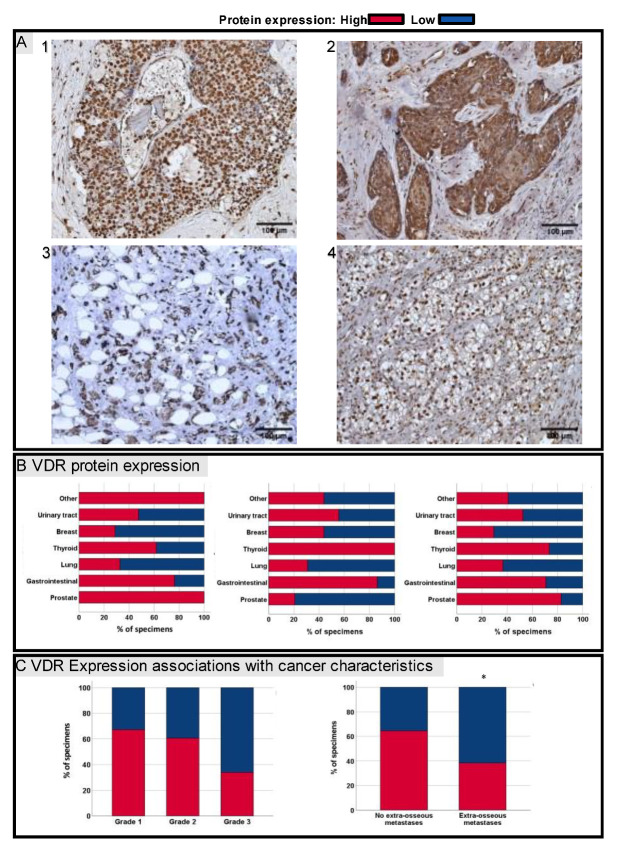
(**A**) Representative images of IHC staining for VDR of bone metastases secondary to prostate (1), oesophagus (2), breast (3) and renal clear cell cancer (4). (**B**) VDR protein expression sorted by primary cancers of bone metastases. Left to right: nucleus, cytoplasm and total VDR protein expression. (**C**) Of the 53 included specimens, 5 were well differentiated, whereas 27 were differentiated as intermediate and 20 as poor. There was a non-significant trend in high-grade cancers towards low nuclear VDR expressions (*p* = 0.07). Additionally, patients with further extra-osseous metastases (multiple metastatic cancers) had reduced nuclear VDR levels compared to patients without other distant metastases (*p* = 0.03); red—high VDR protein expression; blue—low VDR protein expression. Scale bar = 100 µm. * *p* > 0.05.

**Figure 2 jcm-11-06537-f002:**
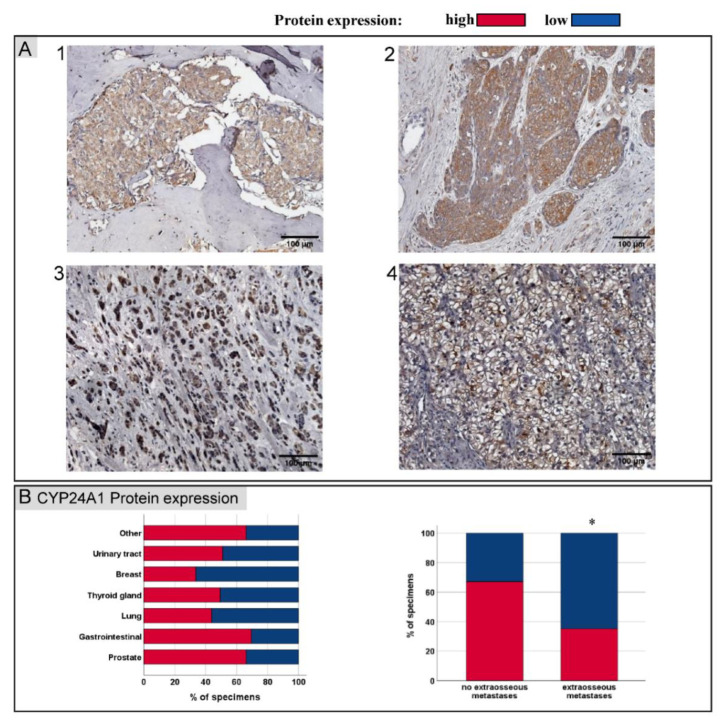
(**A**) Representative images of IHC staining for CYP24A1 of bone metastases corresponding to Figure 1A IHC images secondary to prostate (1), oesophagus- (2), breast- (3) and renal clear cell cancer (4). (**B**) CYP24A1 protein expression sorted by primary cancers of bone metastases. Similar to nuclear VDR protein expression, patients with further extra-osseous metastases (multiple metastatic cancers) had reduced CYP24A1 protein expression compared to patients without other distant metastases (*p* = 0.01); red—high VDR protein expression; blue—low VDR protein expression. Scale bar = 100 µm. * *p* > 0.05.

**Table 1 jcm-11-06537-t001:** Clinical and cancer characteristics of bone metastasis patients with various primary cancers. Two of the four patients in category “other” had oral squamous cell carcinoma, one had a parotid gland carcinoma and one an olfactory neuroblastoma. Abbreviations: T-stage—tumour stage (describes the size/infiltration depth of the primary tumour); N-stage—node stage (describes whether the cancer metastasised to the lymph nodes); Grading, G1—well differentiated—to G3—poorly differentiated (describes the cancer cell differentiation compared to normal cells).

Primary Cancer		T-Stage		N-Stage		Grading		
	** *n* **	**1 to 2**	**3 to 4**	**N0**	**N+**	**G1**	**G2**	**G3**
Breast	16	10 (62.5)	6 (37.5)	5 (31.3)	11 (68.7)	2 (12.5)	6 (37.5)	8 (50.0)
Urinary tract	13	3 (27.3)	8 (62.7)	4 (50.0)	4 (50.0)	2 (15.4)	7 (53.8)	4 (30.7)
Lung	11	1 (12.5)	7 (87.5)	2 (22.2)	7 (77.7)	0 (0)	5 (45.5)	6 (54.5)
Gastro-intestinal	9	2 (25.0)	6 (75.0)	2 (25.0)	6 (75.0)	0 (0)	8 (88.9)	1 (11.1)
Prostate	8	2 (50.0)	2 (50.0)	2 (40.0)	3 (60.0)	Not applied		
Thyroid cancer	5	4 (80.0)	1 (20.0)	1 (100)	0 (0)			
Other	4	0 (0.0)	3 (100)	1 (33.3)	2 (66.6)	1 (25.0)	2 (50.0)	1 (25.0)
Total		22 (40.0)	33 (60.0)	16 (33.3)	32 (66.6)	5 (9.4)	28 (52.8)	20 (37.7)

**Table 2 jcm-11-06537-t002:** Associations between VDR/CYP24A1 protein expression and clinicopathological characteristics. Abbreviations: VDR—vitamin D receptor; No./*n*—number; *—Mann–Whitney U test; ^+^—Pearson’s chi-square test. Bold *p*-values indicate statistical significance (*p* > 0.05).

		VDR Protein Expression			CYP24A1 Protein Expression		
		High	Low	*p*	High	Low	*p*
	*n*	No. of Patients (%)			No. of Patients (%)		
T-Stadium	55			0.57 ^+^			0.32 ^+^
T1–2		13 (59.1)	9 (40.9)		12 (54.5)	10 (45.5)	
T3–4		23 (69.7)	10 (30.3)		20 (60.6)	13 (39.4)	
Grading	53			0.07 ^+^			0.58 ^+^
G1		4 (80.0)	1 (20.0)		1 (20.0)	4 (80.0)	
G2		21 (75.0)	7 (25.0)		16 (57.1)	12 (42.9)	
G3		9 (45.0)	11 (55.0)		11 (55.0)	9 (45.0)	
N-Stadium	48			0.63 ^+^			0.33 ^+^
N0		12 (70.6)	4 (29.4)		12 (70.6)	4 (29.4)	
N+		18 (58.1)	13 (41.9)		16 (50.0)	16 (50.0)	
Multiple metastasised	60			**0.03** ^+^			**0.01** ^+^
Positive		18 (56.3)	14 (43.7)		14 (43.8)	18 (56.2)	
Negative		5 (82.1)	23 (17.9)		22 (78.6)	9 (21.4)	
		Years			Years		
Age at diagnosis	66			0.79 *			0.54 *
Mean		62.1	61.1		61.2	62.7	
Time to metastasis	66			0.41 *			0.68 *
Mean		3.7	2.8		3.6	3.2	

## Data Availability

Not applicable.

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
