# Peer review of "Bone Metastases of Diverse Primary Origin Frequently Express the VDR (Vitamin D Receptor) and CYP24A1"

_jcm, 2022, doi:10.3390/jcm11216537_

Round 1
Reviewer 1 Report
In this study, the authors investigated the expression profiles of VDR and CYP24A1 in various types of human bone metastases by immunohistochemistry. I think this paper carefully described the results, and would contributes to the filed of this research area.
Author Response
We thank the reviewer for the comment and are pleased that the paper was appreciated.
Reviewer 2 Report
The title of the MS is needed to be revised because this type of title have already exist in sites. I am worried about the plagiarism check because parts of the MS have already exist. Is it possible to publish this type of MS.
https://www.thieme-connect.com/products/ejournals/abstract/10.1055/s-0042-1755915
https://www.postersessiononline.eu/173580348_eu/congresos/ECTS2022/aula/-P_79_ECTS2022.pdf
I think that the MS should be revised after deleted the published data.
Reviewer 3 Report
The authors stained vitamin D receptors in bone metastases from various carcinomas with a single antibody and showed that the expression of nuclear vitamin D receptors was qualitatively reduced in cases of high malignancy. However, I did not understand the new finding of the results.
1) Although positive and negative controls for immunostaining are described, please present how they are actually stained. Also, you should describe the specific analysis method of how you calculate the nuclear and cytoplasmic expression levels using them. Just by stating that you referred to a paper, but I cannot determine if this method of evaluation is correct. How exactly the analysis is performed should be expressed using figures, etc.
2) Is it necessary to extract genes and proteins from tissues and compare their expression levels of vitamin D receptor? It is not difficult, since techniques such as bulk RNA-seq are easily established these days. If cost is an issue, then at the very least, qPCR, etc. should be used to find out. Again, I cannot evaluate whether the result of the analysis was correct without following different approach.
3) The statement in the abstract and conclusion that "Vitamin D inoculation is beneficial for patients with bone metastases expressing the vitamin D receptor" is simply restating a known fact. If you want to tie this to the current results, please analyze a sample of patients who were taken to vitamin D and have bone metastases. Or is it possible to separate the tissue samples by how much vitamin D was inoculated? I think it is too much of a leap to conclude that supplements should be taken and from the results of this study.
Minor
There is a non-living CYP24A1 in the abstract. Please at least mention that it is a vitamin D catabolite.
Round 2
Reviewer 3 Report
I have no additional comment.